# A Screening Measure of Emotion Regulation Difficulties: Polish Norms and Psychometrics of the Difficulties in Emotion Regulation Scale-8 (DERS-8)

**DOI:** 10.3390/healthcare13040432

**Published:** 2025-02-18

**Authors:** Paweł Larionow, Monika Mazur, Karolina Mudło-Głagolska

**Affiliations:** 1Faculty of Psychology, Kazimierz Wielki University, 85-064 Bydgoszcz, Poland; mudlo@ukw.edu.pl; 2School of Human Sciences, University of Economics and Human Sciences in Warsaw, 01-043 Warsaw, Poland; monique.mazur@gmail.com

**Keywords:** anxiety, clinical practice, depression, emotion regulation, measurement invariance, norms, psychometric properties, psychopathology, questionnaire, well-being

## Abstract

**Background/Objectives:** Difficulties in emotion regulation (DER) serve as a transdiagnostic risk factor for a wide range of emotion-based psychopathologies, including anxiety and depression disorders. This study presents a report on the psychometrics of the eight-item Difficulties in Emotion Regulation Scale-8 (DERS-8) and the development of its Polish norms. **Methods:** The sample comprised 1329 Polish adults aged 18–73, with 907 females, 384 males, 36 non-binary, and 2 people with an unidentifiable gender. The participants filled out a series of self-report questionnaires on DER, anxiety and depression symptoms, and well-being. Factor structure and measurement invariance, as well as discriminant validity of the DERS-8, were tested with confirmatory factor analysis. Convergent and divergent validity was assessed via relationships with negative and positive mental health outcomes. Internal consistency reliability was evaluated with alpha and omega coefficients. Demographic differences were also examined. **Results:** Our empirical evidence supported the strong psychometrics of the Polish DERS-8, including its good level of internal consistency reliability (i.e., 0.89) and strong validity. The one-factor DERS-8 model had a good fit, with its supported scalar invariance between a set of demographic variables and levels of mental health outcomes. DERS-8 scores were strong positive predictors of anxiety and depression symptoms and negative predictors of well-being, emphasizing the potential detrimental effects of DER on the dual continuum of mental health and mental illness. **Conclusions**: The Polish DERS-8 has strong psychometric properties. Given the development of its percentile rank norms, the scale can be used as a good screening measure of DER in the Polish adult sample.

## 1. Introduction

Emotion regulation is a term used to describe “the processes by which individuals influence which emotions they have, when they have them, and how they experience and express them” [1], (p. 275). Effective emotion regulation is associated with improved health and well-being [2] and allows individuals to foster strong personal and professional relationships [3]. Conversely, difficulties in emotion regulation are found to be a transdiagnostic risk factor for a wide range of psychopathologies [4], including anxiety, depression, substance use, and eating disorders [5]. Given its significance in mental health, accurately measuring difficulties in emotion regulation is crucial for both research and clinical practice.

The emotion regulation field is characterized by a variety of emotion regulation models and their corresponding measures. In a recent systematic review of emotion regulation models [6], ten models were identified. The most prevalent model was the *process model of emotion regulation* by Gross [1,7], with the Emotion Regulation Questionnaire [8] and the Process Model of Emotion Regulation Questionnaire [9] for measuring individual emotion regulation strategies (e.g., expressive suppression). The second most prevalent model was Gratz and Roemer’s model of difficulties in emotion regulation [10]. It identifies four key components of emotion regulation: (1) recognizing and understanding one’s emotions, (2) accepting emotions, (3) managing behavior in alignment with goals while experiencing negative emotional states, and (4) using flexible, context-appropriate emotion regulation strategies to adjust emotional responses as needed to achieve personal goals and meet situational demands [10]. Based on this model, the Difficulties in Emotion Regulation Scale (DERS) was developed [10].

The DERS is one of the commonly used tools for assessing difficulties in emotion regulation (for review, see [11,12]). The original DERS-36 is a self-report questionnaire designed to assess six dimensions of difficulties in emotion regulation: (1) non-acceptance of emotional responses, (2) difficulties engaging in goal-directed behavior, (3) impulse control difficulties, (4) lack of emotional awareness, (5) limited access to emotion regulation strategies, and (6) lack of emotional clarity [10,13]. The DERS-36 is psychometrically reliable [14], making it a cornerstone measure in both research and clinical contexts. Yet, with 36 items, the DERS-36 could be difficult to apply in certain situations or environments, such as large-scale epidemiological research or in a clinical treatment context [14]. The previous evidence suggests that the DERS-36 performed similarly across gender (i.e., females vs. males) and racial groups, supporting meaningful comparisons between demographic groupings [15,16]. Females tended to have greater difficulties in emotion regulation than males; however, the effect size of gender differences was small (e.g., [15,17]).

In response, the DERS-8 was developed to assess difficulties in emotion regulation in both adolescents and adults [18]. This brief, one-dimensional, eight-item self-report scale demonstrates strong psychometric properties [18] while offering advantages in terms of its practical application. The reduced number of items makes the DERS-8 especially valuable in both clinical settings and long-term research supporting targeted interventions for older adults and providing precise emotional difficulties screening [19]. Moreover, given their brevity, short scales are appropriate for studies where participants need to evaluate themselves and others multiple times [20]. Furthermore, in some cases (e.g., online-based research, where respondents are unlikely to spend much time on the website), researchers may encounter constraints that force them to choose between using a very short tool or not measuring at all [20]. Finally, the increasing pace of life, as well as specific research conditions, often forces researchers to work with progressively shorter tools, leading to an increasing need for such measures [21].

Emotion regulation difficulties are suggested to be a transdiagnostic risk factor for a wide range of mental health issues [4]. In Poland, the prevalence of such issues, including high levels of anxiety and depression symptoms as well as decreased levels of well-being, is high [22,23]. Therefore, a brief and effective psychometric tool for emotion regulation difficulties is of high relevance for research and clinical practice, chiefly a screening and prevention of mental health problems. As such, we deemed that the DERS-8 was a good candidate as a screening measure of emotion regulation difficulties in a Polish sample.

Given the above, the aim of the current study was to verify the psychometric properties of the first Polish version of the DERS-8 and to develop its current Polish norms. We hypothesized that

The Polish DERS-8 would be characterized by a one-factor structure, with a good fit in confirmatory factor analysis, and it would be invariant across different demographic categories as well as mental health outcomes levels;The Polish DERS-8 would have good internal consistency reliability;Higher difficulties in emotion regulation, measured with the Polish DERS-8, would be associated with higher levels of psychological distress (i.e., anxiety and depression symptoms) and lower levels of well-being, supporting convergent and divergent validity of the scale;The Polish DERS-8 would show good discriminant validity, with its scores being statistically separable from psychological distress and well-being;The Polish DERS-8 scores would be meaningful statistical predictors of anxiety and depression symptoms as well as psychological well-being while controlling for sociodemographic variables.

Making implications for the assessment and treatment of emotion regulation issues more accessible, we strove to present Polish adult norms for the DERS-8. To help facilitate the interpretation of the scale scores in research and practice, we presented a guide to their interpretation.

## 2. Materials and Methods

### 2.1. Procedure

This research project was conducted in accordance with the Declaration of Helsinki Ethical Principles and was approved by the Ethics Committee of the Faculty of Psychology at Kazimierz Wielki University (No. 1/13.06.2022, with its latest revision: No. 3/11.11.2024). L. Steinberg, one of the authors of the original DERS-8 [18], granted us permission to translate and validate the scale in Polish.

From November to December 2024, participants were invited to complete our online anonymous survey on the social media platforms Facebook and Instagram, where we posted a link to the study with an appended consent form. This link was distributed on the authors’ profiles. The survey was hosted on the Google Forms platform. All respondents digitally provided their written informed consent before the survey completion. There were no missing data, as replies on all questionnaires were mandatory.

The participants were people from a Polish adult population without any specific features that were targeted during the recruitment process. The inclusion criteria were Polish-speaking people and an age of 18 years and above, who signed their informed consent for study participation. The exclusion criterion was inattentive responding (such respondents failed an attention check question that requested them to select a specific answer). A total of 1388 people replied to the consent form, and the data of 59 people were considered invalid as specified by the inclusion and exclusion criteria. As such, the final sample comprised of 1329 people.

### 2.2. Participants

The sample comprised 1329 Polish-speaking adults (907 females, 384 males, 36 non-binary, and 2 participants who did not report their gender) with ages ranging from 18 to 73 years. Detailed demographic characteristics are displayed in Table 1.

### 2.3. Measures

All participants filled out a demographic form and a series of self-report questionnaires. One of these tools is not described in this study, as it was devoted to another ongoing and unpublished research project.

In this study, all questionnaires were used in their Polish versions, which previously demonstrated strong psychometric properties. Internal consistency reliability coefficients obtained in the current study were good, with omega and alpha coefficients of ≥0.79 (see Section 3.1 for details). Further measure descriptions are provided below.

#### 2.3.1. The Demographic Questionnaire

A demographic form consisted of questions on age, gender, area of residence, education level, and relationship status.

#### 2.3.2. The Difficulties in Emotion Regulation Scale-8 (DERS-8) and Its Translation

The DERS-8 is a brief self-report measure designed to assess difficulties in emotion regulation in both adolescents and adults [18]. This unifactorial scale consists of eight items. Each item starts with “When I’m upset” (e.g., “When I’m upset, I have difficulty getting work done”) to ensure respondents focus on situations requiring emotion regulation of negative emotions. Items are scored on a 5-point scale, ranging from 1 (almost never, 0–10%) to 5 (almost always, 91–100%). The total DERS-8 score is obtained by summing the individual item scores, with higher scores indicating more pronounced difficulties with emotion regulation.

To create the Polish version of the DERS-8, we adhered to a standard translation process [24]. The initial step involved translating the English version of the DERS-8 into Polish, which was carried out by three independent translators. These translations were then merged into a single Polish version. Subsequently, an independent translator conducted a back translation into English. This version was reviewed and compared with the original English version. Following this review, minor refinements were made, resulting in the finalized Polish version of the DERS-8. This final scale, administered in this study, is posted in the Appendix A.

#### 2.3.3. The Patient Health Questionnaire-4 (PHQ-4)

The PHQ-4 is a brief screening questionnaire used to detect anxiety and depression symptoms over the previous two weeks [25,26]. The PHQ-4 consists of four items and two subscales: Anxiety and Depression. Each subscale contains two items. Examples of the items are “Feeling nervous, anxious, or on edge” for the Anxiety subscale and “Feeling down, depressed, or hopeless” for the Depression subscale. Moreover, a total score representing an overall marker of psychological distress can be computed. PHQ-4 items are scored on a 4-point scale from 0 (not at all) to 3 (nearly every day), with higher scores indicating higher levels of symptoms. A total score of ≥6 indicates elevated levels of psychological distress and suggests probable anxiety and/or depressive disorder. In this study, the Polish version of the PHQ-4 was used [23]. The Polish PHQ-4 has previously demonstrated strong psychometric properties, including an empirically supported 2-factor structure and good internal consistency reliability of ≥0.73 in a general community sample of Polish females and males [23].

#### 2.3.4. The WHO-Five Well-Being Index (WHO-5)

The WHO-5 is a brief self-report questionnaire for measuring positive well-being [27,28,29]. The WHO-5 consists of five items (e.g., “I feel cheerful and in good spirits”), which are scored on a 6-point scale ranging from 0 (at no time) to 5 (all the time). Higher scores indicate a higher level of well-being. The WHO-5 can act as a screening measure of depression, with scores of <13 serving as a cut-off for mental ill-being. As such, scores from 0 to 12 indicate a possible presence of mental illness (particularly depressive disorder) and suggest a need for further comprehensive mental health assessments [28]. In this study, the Polish WHO-5, which demonstrated good psychometric properties in both general community and clinical samples, was used [22,30]. For instance, this Polish tool was characterized by an empirically supported 1-factor structure and internal consistency reliability of 0.85 in a general community sample of Poles [22].

### 2.4. Analytic Strategy

#### 2.4.1. Factor Structure and Measurement Invariance

Confirmatory factor analysis was carried out using *R* v. 4.4.2 with the *lavaan* statistical package, and *JASP* v. 0.19.2 was used for all other analyses. For factor analytic studies, a sample size of more than 1000 participants is generally treated as excellent [31]. As such, our sample size was appropriate for the examination of the DERS-8.

Confirmatory factor analysis with the diagonally weighted least squares estimator was applied. A theoretically informed 1-factor model of the DERS-8 was tested. In this model, all eight items were specified to load on a general emotion regulation difficulties factor. Goodness-of-fit was judged based on the following fit index values: root mean square error of approximation (RMSEA) with 90% confidence intervals (90% CI), standardized root mean square residual (SRMR), comparative fit index (CFI), and Tucker–Lewis index (TLI). RMSEA and SRMR values ≤ 0.08 indicate acceptable fit, and values ≤0.06 indicate excellent fit. CFI and TLI values ≥ 0.90 indicate acceptable fit, and values ≥ 0.95 indicate excellent fit [32].

The measurement invariance [33] of the DERS-8 was examined on the configural, metric, and scalar levels across different demographic categories and levels of mental health outcomes. In the measurement invariance testing, the demographic categories were (1) two genders (females vs. males), (2) two age groups (younger people aged 18–29 vs. older people aged 30–73), (3) two education categories (no university degree vs. university degree), and (4) two relationship status groups (single vs. in relationships). The mental health outcomes categories were (1) high vs. low psychological distress symptoms groups (based on the PHQ-4 score classification, where a total PHQ-4 of ≥6 indicates high anxiety and depression symptoms) and (2) low well-being vs. normal well-being levels (based on the WHO-5 score classification, where a total WHO-5 score of ≤12 indicates low well-being). The configural, metric, and scalar invariance models were compared in terms of the CFI when an absolute difference in CFI (ΔCFI) of <0.01 indicates full invariance [34].

#### 2.4.2. Internal Consistency Reliability

McDonald’s omega values and Cronbach’s alpha coefficients with 95% confidence intervals (95% CI) were calculated. For these coefficients, values ≥ 0.70 were judged as acceptable, ≥0.80 as good, and ≥0.90 as excellent [35].

#### 2.4.3. Convergent and Divergent Validity

To assess the convergent and divergent validity of the DERS-8, we computed Pearson correlations between DERS-8 scores, anxiety and depression symptoms (as measured with the PHQ-4), and well-being (as measured with the WHO-5).

#### 2.4.4. Discriminant Validity

The discriminant validity of the DERS-8 was assessed against other negative (i.e., anxiety and depression symptoms) and positive mental health (i.e., well-being) outcomes. We tested and compared three models: Model 1, where all items of all measures (i.e., the DERS-8, PHQ-4, and WHO-5) loaded onto the single mental health factor; Model 2 was the 4-factor uncorrelated model (the DERS-8 score, PHQ-4 Anxiety score, PHQ-4 Depression score, and WHO-5 score were the four uncorrelated factors); Model 3 was the 4-factor correlated model (the DERS-8 score, PHQ-4 Anxiety score, PHQ-4 Depression score, and WHO-5 score were the four correlated factors). We predicted that Model 3 would have the best fit across these three models, suggesting that emotion regulation difficulties, anxiety and depression symptoms, and well-being are related but distinct psychological phenomena (see [36]). The goodness-of-fit of these models was assessed based on the same criteria as used for the confirmatory factor analysis (see Section 2.4.1).

Additionally, we evaluated the 95% CI for the estimated correlations between the constructs from confirmatory factor analysis. If the 95% CI for the estimated correlation between the two constructs does not include 1.0, this supports evidence of discriminant validity for these two constructs [37]. Finally, heterotrait–monotrait (HTMT) ratio of correlations (with a threshold value of 0.85) and an average variance extracted (AVE) value (with a threshold value of 0.5) were used to examine discriminant validity [37].

#### 2.4.5. Predictive Ability

A series of multiple hierarchical regression analyses were conducted to assess whether emotion regulation difficulties, as measured with the DERS-8, were a statistical predictor of anxiety and depression symptoms and well-being. As such, in our regression analyses, the DERS-8 scores were treated as a predictor, whereas the PHQ-4 Total score and WHO-5 Total score were treated as the dependent variables. Also, we controlled for the potential effects of demographic variables (i.e., gender, age, residence, education, and relationship status) on mental ill-being and well-being. The demographic variables were introduced into the regression models in Step 1, whereas the DERS-8 scores were input into the regression models in Step 2. To prevent potential multicollinearity issues, we assessed the tolerance values for each predictor, with values of ≤0.10 indicating multicollinearity [38]. Influential cases (outliers) were assessed using Cook’s distance with a cut-off of 1. No outliers were detected.

#### 2.4.6. Demographic Comparisons

Pearson correlations between age and DERS-8 scores were calculated. To compare DERS-8 scores between people from different demographic categories, an analysis of covariance (ANCOVA), with age used as a covariate, was conducted.

#### 2.4.7. Development of Polish Norms

Based on the approach described in the literature [39], Bayesian percentile rank norms with their corresponding 90% and 95% credible intervals were calculated for the total sample (*n* = 1329) and two age groups (i.e., 18–29 and 30–73 years old).

## 3. Results

### 3.1. Descriptive Analysis and Internal Consistency Reliability

Table 2 demonstrates descriptive statistics of the study variables across gender groups. All the study variables were reasonably normally distributed in the total sample (*n* = 1329), with skewness values from 0.12 to 0.55 and kurtosis values from −1.11 to −0.34.

Internal consistency reliability of the DERS-8 was good, with McDonald’s Omega of 0.88 and Cronbach’s Alpha of 0.89. The other questionnaires also showed good reliability in this study, with omega and alpha coefficients of ≥0.79.

### 3.2. Factor Structure and Its Invariance

Descriptive statistics for the DERS-8 items are displayed in Table 3.

The intended one-factor DERS-8 model had a good fit to the data (e.g., CFI = 0.986, RMSEA = 0.072, SRMR = 0.056, see Table 4). All factor loadings were high (≥0.601, all *ps* < 0.001, see Table 3).

The one-factor DERS-8 model showed configural, metric, and scalar level invariance across all the tested demographic categories and psychological distress (PHQ-4) and well-being (WHO-5) levels, with the absolute difference of CFI being equal to 0.009 and less between the invariance levels (see Table 4).

### 3.3. Convergent and Divergent Validity

As expected, DERS-8 scores were strongly positively associated with anxiety and depression symptoms and negatively with well-being (see Table 5), supporting convergent and divergent validity.

### 3.4. Discriminant Validity

We conducted two series of confirmatory factor analyses to examine the discriminant validity of the DERS-8. We were interested in whether difficulties in emotion regulation were a statistically separable construct from anxiety and depression symptoms as well as well-being (see Figure 1 with the tested models).

As expected, the single mental health factor model had a poor fit to the data. The four-factor model with uncorrelated (orthogonal) factors was not identified (see Table 6); therefore, factor loadings could not be computed.

In contrast, the four-factor correlated model, with the separate DERS-8, PHQ-4 Anxiety, PHQ-4 Depression, and WHO-5 factors, had an excellent fit to the data. Therefore, our analysis showed that difficulties in emotion regulation were distinct from psychological distress and well-being. Additionally, the 95% CI for the estimated correlations between all the pairs of the examined constructs did not include 1.0, supporting evidence of discriminant validity for the constructs. Finally, AVE values were ≥0.57, and values of HTMT ratio of correlations were ≤0.81, further supporting good discriminant validity of the four constructs, as measured with the DERS-8, two PHQ-4 subscale scores, and WHO-5.

### 3.5. Predicting Mental Health Outcomes Based on DERS-8 Scores

The regression analysis demonstrated that DERS-8 scores, adjusting for demographic variables, were a strong positive predictor of anxiety and depression symptoms (beta = 0.60, *p* < 0.001) and a strong negative predictor of well-being (beta = −0.47, *p* < 0.001). Detailed results are displayed in Table 7 and Table 8.

#### 3.5.1. Predicting Psychological Distress Based on DERS-8 Scores

In Step 2, among demographic variables, only relationship status was a statistically significant predictor of anxiety and depression symptoms, with being in relationships associated with less emotion regulation difficulties (see Table 7). Beyond demographic variables, the DERS-8 scores explained 32.40% of the variance of anxiety and depression symptoms (Δ adjusted R^2^ between Step 2 and Step 1).

#### 3.5.2. Predicting Well-Being Based on DERS-8 Scores

In Step 2, among demographic variables, only relationship status was a statistically significant predictor of well-being, with being in relationships associated with higher levels of well-being (see Table 8). Beyond demographic variables, the DERS-8 scores explained 19.44% of the variance of well-being (Δ adjusted R^2^ between Step 2 and Step 1). Overall, our regression analyses showed that emotion regulation difficulties could statistically significantly predict negative and positive mental health outcomes.

### 3.6. Demographic Differences

We used three series of ANCOVAs (with age used as a covariate) to compare DERS-8 scores between (1) females and males, (2) people without university degrees and people with university degrees, and (3) singles and people in relationships. There were gender differences in emotion regulation difficulties, with females tending to have higher DERS-8 scores than males, *F*_(1, 1288)_ = 28.24, *p* < 0.001, η^2^ = 0.02. People without university degrees tended to have higher DERS-8 scores than people with university degrees, *F*_(1, 1326)_ = 4.79, *p* = 0.029, η^2^ = 0.003. Relationship status did not differentiate DERS-8 scores (*p* < 0.05). In all these three series of ANCOVAs, age was a statistically significant covariate. Pearson correlations demonstrated that age was negatively correlated with DERS-8 scores in females (*r* = −0.29, *p* < 0.001) and males (*r* = −0.28, *p* < 0.001), suggesting that younger people tended to have more emotion regulation difficulties.

### 3.7. Group Norms

We calculated group norms for the DERS-8 scores across the total sample and, due to age differences in emotion regulation difficulties, for two age groups (i.e., aged 18–29 and 30–73) separately (see Appendix A).

## 4. Discussion

The aim of the study was to examine the psychometric properties of the Polish version of the DERS-8 and to present its norms. Overall, much like the original English version [18], the Polish DERS-8 showed strong psychometric performance, including factorial validity, internal consistency reliability, and convergent, divergent, and discriminant validity.

### 4.1. Factor Structure and Measurement Invariance

As expected, the one-factor structure of the Polish DERS-8 was supported empirically in confirmatory factor analysis. The scale was also characterized by strict invariance across different demographic categories, including gender, age, education, and relationship status, supporting the possibility of comparing the mean DERS-8 scores between these demographic groups meaningfully. As a further contribution to the emotion regulation field, we also tested whether the DERS-8 structure was the same between people with different levels of mental health outcomes (i.e., between people with normal and high psychological distress levels and between people with normal and low well-being levels). In this case, our analysis also supported the strict invariance of the DERS-8, suggesting that regardless of different levels of mental health outcomes, difficulties in emotion regulation could be effectively measured with the Polish DERS-8. Our results are in line with the conclusions presented in the original DERS-8 paper [18], where the scale showed strong replicability and clinical relevance in assessing emotion regulation difficulties between clinical and non-clinical samples. Taken together, the current empirical evidence suggests that the DERS-8 has demonstrated strong factorial validity across Penner et al.’s study [18] and this Polish one. This evidence is promising; however, to the best authors’ knowledge, the Polish version of the scale seems to be the first non-English translated version of the tool. Therefore, more studies are required to investigate its factor structure across different cultural contexts.

### 4.2. Internal Consistency Reliability and Convergent, Divergent and Discriminant Validity

Consistent with the original DERS-8 validation study [18], the internal consistency reliability of the total scale scores, as measured with McDonald’s Omega and Cronbach’s Alpha, was good. This indicates that the measurement of emotion regulation difficulties with the DERS-8 is robust, despite the tool’s brevity.

In terms of convergent and divergent validity, Penner et al. [18] noted reasonable positive correlations between the DERS-8 scores and other negative mental health outcomes (e.g., borderline personality, internalizing and externalizing problems, etc.), somatic complaints, and neuroticism from the Big Five personality. In our study, we were interested in not only exploring whether emotion regulation difficulties were related to negative mental health outcomes but also whether these difficulties were related to a decline in well-being, serving as one of the positive mental health outcomes. In our Polish study with its bivariate correlational analysis, DERS-8 scores were strongly positively related to anxiety and depression symptoms and negatively related to well-being. The DERS-8 scores were more strongly associated with anxiety symptoms (*r* = 0.62) than depression ones (*r* = 0.51) and similarly strongly but negatively associated with well-being (*r* = −0.48).

Despite these strong correlations between the constructs of interest (i.e., emotion regulation difficulties, psychological distress, and well-being), our discriminant validity analysis indicated that emotion regulation difficulties were statistically separable from psychological distress and well-being scores, as it was shown in a series of confirmatory factor analyses. The emotion regulation difficulties construct did not overlap with the psychological distress construct and the well-being one, suggesting that the DERS-8 scores could serve as an independent statistical predictor of other negative and positive mental health outcomes.

### 4.3. Predictive Ability of the DERS-8 Scores

We conducted two series of regression analyses and noted that emotion regulation difficulties (controlling for other demographic variables) were significant predictors of psychological distress (i.e., anxiety and depression symptoms) and well-being scores. Our research data supported the idea that these difficulties could be associated with both negative and positive mental health outcomes, suggesting that targeting emotion regulation difficulties might be beneficial for decreasing mental health burden and improving well-being. These difficulties were somewhat more strongly associated with psychological distress than with well-being, indicating a specific pattern across these links.

Overall, our results support past evidence regarding strong positive correlations of emotion regulation difficulties, measured with the DERS-8 and its long forms, with anxiety and depressive symptoms (e.g., [13,40]). We deem that the assessment of emotion-related constructs should include the perspective of the two continua model of mental health [41], with its mental illness and mental well-being dimensions. In such cases, both negative and positive mental health outcomes are treated as potential correlates of the assessed constructs. In our study, taking into account both strong positive and negative correlations of emotion regulation difficulties with anxiety and depression symptoms and well-being scores, respectively, DERS-8 scores might serve as a potential transdiagnostic risk factor for common psychopathologies and poor well-being. However, causality should be determined in prospective studies.

### 4.4. Demographic Differences in Emotion Regulation Difficulties

Our study revealed several demographic patterns in emotion regulation difficulties. Females reported greater struggles with managing negative emotions compared to males, supporting previous empirical findings that emotional challenges may be more pronounced in females [42,43]. However, the effect size of these gender differences was small, which is consistent with the previous findings [44].

In a series of ANCOVAs devoted to the demographic differences analysis, age was a significant covariate. Further exploration of the role of age in emotion regulation difficulties allowed us to note that difficulty regulating emotions was more prevalent among younger individuals than in the older ones in our data-set. These results are consistent with earlier research, suggesting that aging is associated with more efficient and adaptive emotion regulation due to the accumulation of life experience [45,46]. Our results indicate that younger individuals are at risk of psychopathology development, therefore assessing emotion regulation difficulties, which serve as a risk factor for a wide range of psychopathologies [47,48,49], is particularly relevant for this group.

We were also interested in discovering whether education and relationship status categories (as less studied demographic variables in the emotion regulation field) differentiated difficulties with emotion regulation. We found that people without a university degree experienced slightly more difficulties in emotion regulation, but the effect size of these differences was negligible. The relationship status did not differentiate the DERS-8 scores.

To date, to the best of our knowledge, there are no studies on the demographic differences in DERS-8 scores, therefore, we cannot contrast our results with the previous ones. However, based on the obtained results and the past work on the full DERS (e.g., [44]), we may conclude that age and gender seem to be the most relevant demographic factors, which should be taken into account while studying challenges in emotion regulation.

### 4.5. Norms and Their Interpretation and Other Practical Implications

Based on our data-set, we calculated Polish norms of the DERS-8 for the total sample and two age groups (aged 18–29 and 30–73 years old). The percentile rank norms were computed using a Bayesian approach, with 90% and 95% credible intervals for percentile rank scores. To facilitate the interpretation of the DERS-8 scores, we proposed to classify levels of emotion regulation difficulties based on the following categories of percentile rank norms [50]: low levels (percentile ranks of ≤15), average levels (percentile ranks from 16 to 84), and high levels (percentile ranks of ≥85). Overall, a total DERS-8 score of ≥32 suggested high levels of emotion regulation difficulties in the total sample. We recommend using age-specific cut-off scores for high levels of these difficulties, with a total DERS-8 score of ≥33 in younger adults aged 18–29 and a total DERS-8 score of ≥29 in older adults aged 30–73. We would like to underline that these cut-off scores were calculated based on the current Polish sample and, therefore, may not generalize to other cultures.

The growing pace of life, along with specific research conditions, frequently requires researchers to use increasingly shorter tools, resulting in a greater demand for such measures [21]. The concise design of the DERS-8 makes the assessment of difficulties in emotion regulation more practical and accessible in diverse settings. For instance, given its reduced number of items, DERS-8 is a good measure for quick screenings in general and clinical populations.

The development and implementation of interventions targeting emotion regulation in children and adults requires the use of well-operationalized measures of emotion regulation, with the DERS being a popular measure of interest [11]. In the educational context, following the ideas that emotion regulation difficulties are positively related to academic burnout [51] and student engagement and relations with peers and teachers [52], we believe that the DERS-8 can be considered a good measure for quick screening of emotion regulation difficulties among students. In Poland, university students experienced high levels of anxiety and depression symptoms [53]; therefore, screening assessments of emotion regulation difficulties and the development of psychological support programs targeting psychological distress and emotion regulation abilities seem to be pertinent. With the brevity of the DERS-8, such monitoring is very accessible.

Additionally, the DERS-8 supports long-term research aimed at targeted interventions for older adults [19]. Moreover, the DERS-8 is also well-suited for studies requiring the tracking of emotion regulation difficulties over time or simultaneous evaluation of many constructs.

### 4.6. Limitations

This study has several limitations. First, its cross-sectional design does not allow for determining whether emotion regulation difficulties lead to mental health issues or vice versa, highlighting the need for further investigation of causality through longitudinal research. Secondly, our sample was not fully representative of the structure of the entire Polish population (e.g., the presence of gender imbalance). Also, participants choose to engage in psychological studies that align with their personal needs and traits, potentially introducing an unintentional self-selection bias [54]. Additionally, online recruitment could have excluded individuals with limited Internet access, potentially leading to underrepresentation of certain groups (e.g., older individuals and people with limited Internet access). Thirdly, the small number of non-binary participants limited the ability to explore gender-based differences fully between females, males, and non-binary individuals, as well as to conduct measurement invariance across these three gender groups. Finally, by design, this was not a clinical study; therefore, we did not control participants’ clinical diagnoses, if there were any. Our study focused on a non-clinical adult sample, offering a foundation for future research involving more diverse groups, such as adolescents and individuals with various clinical diagnoses.

### 4.7. Future Directions

Several areas should be explored in future research on the DERS-8. First, while the DERS-8 primarily focuses on emotion regulation difficulties related to negative emotions, as the items begin with “When I am upset” [18], it is relevant to develop a short tool that also assesses the difficulties in regulation of positive emotions [55,56]. Secondly, the DERS-8 was created to study both adults and adolescents [18]; however, this study focused solely on adults. Consequently, future research should include adolescent populations to assess the scale’s validity and applicability across age groups. We also feel that there is an urge to study emotion-related variables in people with genders other than binary ones (i.e., female and male) to increase their visibility in the research and practice of healthcare [57,58]. Finally, as this study did not include clinical samples, future research should examine the DERS-8 among individuals with clinical diagnoses to evaluate its relevance and utility in clinical settings (e.g., [13]).

## 5. Conclusions

The DERS-8 has demonstrated strong psychometric properties in an adult Polish sample. The Polish DERS-8 was characterized by the theoretically informed one-factor structure, which was invariant across several demographic categories (i.e., gender, age, education, and relationship status) and different levels of mental health outcomes. This indicates that DERS-8 scores can be collated across these groups pertinently. A good level of internal consistency reliability and strong convergent and divergent validity further supported the good psychometric performance of the scale. Moreover, emotion regulation difficulties, as measured with the DERS-8, were statistically set apart from the current level of one’s psychological distress, indicating that these constructs were theoretically and empirically distinguishable. As such, DERS-8 scores were strong predictors of anxiety and depression symptoms and well-being. As we calculated the current Polish norms for the freely available DERS-8, we believe that our study contributes to the clinical practice well, creating opportunities for the efficient prevention of emotion-based psychopathology among Polish adults.

## Figures and Tables

**Figure 1 healthcare-13-00432-f001:**
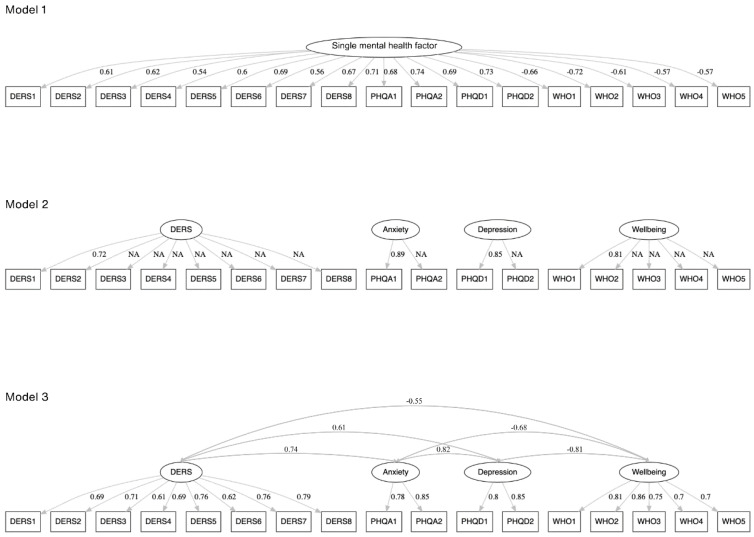
The confirmatory factor analysis models of the mental health indicators (*n* = 1329). *Note*. NA = Not available. Standardized factor loadings are shown (Models 1 and 3). Factor loadings in Model 2 could not be computed, as this model was not identified. Estimated correlations between factors are displayed (Model 3).

**Table 1 healthcare-13-00432-t001:** Demographic characteristics of the study sample.

Demographic Categories		*n*	%
Age	*M* = 28.67, *SD* = 12.03, min. = 18, max. = 73	1329	100
Gender	Females	907	68.25
	Males	384	28.89
	Non-binary	36	2.71
	Not reported/unidentifiable	2	0.15
Area of residence	Large cities (above 100,000 inhabitants)	588	44.24
	Towns (from 20,000 to 100,000)	273	20.54
	Small towns (up to 20,000)	144	10.84
	Villages	324	24.38
Education	Higher	536	40.33
	Secondary	674	50.71
	Vocational	52	3.91
	Primary	67	5.04
Relationship status	Single	612	46.05
	In relationships	717	53.95

**Table 2 healthcare-13-00432-t002:** Descriptive statistics for the study variables.

Variables	Total Sample (*n* = 1329)				Females (*n* = 907)		Males (*n* = 384)		Non-Binary (*n* = 36)	
	McDonald’s Omega (95% CI)	Cronbach’s Alpha (95% CI)	*M*	*SD*	*M*	*SD*	*M*	*SD*	*M*	*SD*
DERS-8 Total score	0.88 (0.88; 0.89)	0.89 (0.88; 0.89)	22.93	7.57	23.59	7.37	21.08	7.68	26	7.99
PHQ-4 Anxiety	0.79 (0.77; 0.81)	0.79 (0.77; 0.81)	3.18	1.81	3.32	1.79	2.8	1.81	3.61	1.76
PHQ-4 Depression	0.81 (0.79; 0.83)	0.81 (0.79; 0.83)	2.81	1.94	2.82	1.93	2.76	1.96	3.28	1.91
PHQ-4 Total score	0.85 (0.84; 0.87)	0.85 (0.84; 0.86)	5.99	3.41	6.13	3.42	5.57	3.37	6.89	3.27
WHO-5 Total score	0.87 (0.86; 0.89)	0.88 (0.86; 0.89)	8.72	4.92	8.61	4.84	9.05	5.15	7.92	4.33

**Table 3 healthcare-13-00432-t003:** Descriptive statistics of the DERS-8 items and standardized factor loadings from confirmatory factor analysis of the 1-factor model (n = 1329).

DERS-8 Items	*M*	*SD*	Skewness	Kurtosis	Factor Loadings
1. When I’m upset, I have difficulty getting work done.	3.01	1.18	0.12	−1.02	0.72
2. When I’m upset, I feel out of control.	2.5	1.24	0.47	−0.87	0.755
3. When I’m upset, I feel ashamed with myself for feeling that way.	2.34	1.38	0.7	−0.82	0.601
4. When I’m upset, I have difficulty controlling my behaviors.	2.46	1.23	0.59	−0.66	0.718
5. When I’m upset, I believe that there is nothing I can do to make myself feel better.	2.72	1.3	0.34	−1.03	0.699
6. When I’m upset, I become irritated with myself for feeling that way.	3.32	1.36	−0.25	−1.21	0.617
7. When I’m upset, I have difficulty thinking about anything else.	3.4	1.24	−0.22	−1.13	0.774
8. When I’m upset, it takes me a long time to feel better.	3.18	1.23	−0.03	−1.05	0.746

**Table 4 healthcare-13-00432-t004:** Factor structure and measurement invariance for the 1-factor DERS-8 model.

Samples	χ2 (df)	CFI	TLI	RMSEA (90% CI)	SRMR	ΔCFI	Invariance Testing Results
Total sample (*n* = 1329)	158.365 (20)	0.986	0.98	0.072 (0.062; 0.083)	0.056	–	–
Gender invariance (females [*n* = 907] vs. males [*n* = 384])							
Configural	157.545 (40)	0.987	0.982	0.068 (0.057; 0.079)	0.052	–	–
Metric	177.218 (47)	0.986	0.983	0.066 (0.055; 0.076)	0.055	−0.001	Supported
Scalar	196.126 (54)	0.984	0.984	0.064 (0.054; 0.074)	0.058	−0.002	Supported
Age invariance (younger people aged 18–29 [*n* = 900] vs. older people aged 30–73 [*n* = 429])							
Configural	171.707 (40)	0.985	0.979	0.070 (0.060; 0.081)	0.054	–	–
Metric	207.280 (47)	0.982	0.978	0.072 (0.062; 0.082)	0.059	−0.003	Supported
Scalar	225.966 (54)	0.98	0.979	0.069 (0.060; 0.079)	0.062	−0.002	Supported
Education levels (no university degree [*n* = 793] vs. university degree [*n* = 536])							
Configural	161.340 (40)	0.987	0.981	0.068 (0.057; 0.079)	0.053	–	–
Metric	190.531 (47)	0.984	0.981	0.068 (0.058; 0.078)	0.056	−0.003	Supported
Scalar	198.051 (54)	0.984	0.984	0.063 (0.054; 0.073)	0.057	0	Supported
Relationship status (single [*n* = 612] vs. in relationships [*n* = 717])							
Configural	167.447 (40)	0.987	0.982	0.069 (0.059; 0.080)	0.052	–	–
Metric	176.085 (47)	0.987	0.984	0.064 (0.054; 0.075)	0.054	0	Supported
Scalar	179.516 (54)	0.987	0.987	0.059 (0.050; 0.069)	0.054	0	Supported
Psychological distress levels (PHQ-4 Total score of 0–5 [*n* = 644] vs. PHQ-4 Total score of 6–12 [*n* = 685])							
Configural	186.194 (40)	0.976	0.966	0.074 (0.064; 0.085)	0.059	–	–
Metric	247.764 (47)	0.967	0.961	0.080 (0.071; 0.090)	0.066	−0.009	Supported
Scalar	287.450 (54)	0.962	0.96	0.081 (0.072; 0.090)	0.071	−0.005	Supported
Well-being levels (WHO-5 Total score of 0–12 [*n* = 1023] vs. WHO-5 Total score of 13–25 [*n* = 306])							
Configural	176.164 (40)	0.982	0.974	0.072 (0.061; 0.083)	0.056	–	–
Metric	214.764 (47)	0.977	0.973	0.073 (0.064; 0.083)	0.061	−0.005	Supported
Scalar	243.192 (54)	0.975	0.974	0.073 (0.064; 0.082)	0.065	−0.002	Supported

**Table 5 healthcare-13-00432-t005:** Pearson correlations between the study variables (*n* = 1329).

Variables	DERS-8 Total Score	PHQ-4 Anxiety	PHQ-4 Depression	PHQ-4 Total Score	WHO-5 Total Score
DERS-8 Total score	—				
PHQ-4 Anxiety	0.62	—			
PHQ-4 Depression	0.51	0.66	—		
PHQ-4 Total score	0.62	0.9	0.92	—	
WHO-5 Total score	−0.48	−0.56	−0.68	−0.69	—

*Note*. All correlation coefficients are statistically significant (*ps* < 0.001).

**Table 6 healthcare-13-00432-t006:** The confirmatory factor analysis results of the different structural models of mental health indicators (*n* = 1329).

Models	Model Descriptions	χ2 (df)	CFI	TLI	RMSEA (90% CI)	SRMR
Model 1	All items of all measures (i.e., the DERS-8, PHQ-4, and WHO-5) loaded onto the single mental health factor	2113.840 (119)	0.942	0.934	0.112 (0.108; 0.117)	0.103
Model 2 *	The 4-factor model, with the DERS-8 score, PHQ-4 Anxiety score, PHQ-4 Depression score, and WHO-5 score as the four uncorrelated (orthogonal) factors	19,821.100 (119)	0.431	0.35	0.353 (0.349; 0.357)	0.306
Model 3	The 4-factor correlated model, with the DERS-8 score, PHQ-4 Anxiety score, PHQ-4 Depression score, and WHO-5 score as the four correlated factors	418.887 (113)	0.991	0.989	0.045 (0.041: 0.050)	0.045

*Note*. * Model 2 was not identified (the information matrix could not be inverted), suggesting model’s problems; therefore, factor loadings could not be computed.

**Table 7 healthcare-13-00432-t007:** Regression analysis of prediction of psychological distress measured with the PHQ-4 Total score (*n* = 1291).

Model	Predictors	Unstandardized	Standard Error	Standardized (beta)	*t*	*p*	Tolerance
Step 1	(Intercept)	10.21	0.58	–	17.72	<0.001	–
	**Gender**	**−0.63**	**0.2**	**−0.09**	**−3.11**	**0.002**	**0.96**
	**Age**	**−0.05**	**0.01**	**−0.19**	**−6.35**	**<0.001**	**0.8**
	Residence	−0.04	0.08	−0.01	−0.51	0.611	0.97
	Education	−0.23	0.14	−0.05	−1.71	0.087	0.8
	**Relationship status**	**−0.65**	**0.19**	**−0.09**	**−3.4**	**<0.001**	**0.93**
Step 2	(Intercept)	1.21	0.58	–	2.1	0.036	–
	Gender	0.01	0.17	0	0.08	0.937	0.94
	Age	−0.01	0.01	−0.04	−1.4	0.161	0.75
	Residence	0.02	0.06	0.01	0.38	0.703	0.97
	Education	−0.12	0.11	−0.03	−1.07	0.285	0.79
	**Relationship status**	**−0.57**	**0.15**	**−0.08**	**−3.69**	**<0.001**	**0.93**
	**DERS-8 Total score**	**0.27**	**0.01**	**0.6**	**26.19**	**<0.001**	**0.89**

*Note*. Step 1: *F*(5, 1285) = 19.84, *p* < 0.001, adjusted R^2^ = 6.81%. Step 2: *F*(6, 1284) = 139.67, *p* < 0.001, adjusted R^2^ = 39.21%. Gender was coded as following: 1 = females, 2 = males. Residence was coded as following: 1 = villages, 2 = small towns (up to 20,000), 3 = towns (from 20,000 to 100,000), 4 = large cities (above 100,000 inhabitants). Education was coded as following: 1 = primary, 2 = vocational, 3 = secondary, 4 = higher. Relationship status was coded as following: 1 = single, 2 = in relationships. Statistically significant predictors are in bold. In our regression analysis, we controlled for gender, and for interpretability reasons data on 36 non-binary and 2 people with unidentifiable gender were excluded from the analysis; therefore, in this analysis, our sample comprised females and males only and it consisted of 1291 people.

**Table 8 healthcare-13-00432-t008:** Regression analysis of prediction of well-being measured with the WHO-5 Total score (*n* = 1291).

Model	Predictors	Unstandardized	Standard Error	Standardized (beta)	*t*	*p*	Tolerance
Step 1	(Intercept)	3.87	0.85	–	4.58	<0.001	–
	Gender	0.56	0.3	0.05	1.88	0.06	0.96
	**Age**	**0.06**	**0.01**	**0.15**	**4.76**	**<0.001**	**0.8**
	Residence	0.01	0.11	0	0.06	0.956	0.97
	Education	0.31	0.2	0.05	1.54	0.123	0.8
	**Relationship status**	**0.91**	**0.28**	**0.09**	**3.25**	**0.001**	**0.93**
Step 2	(Intercept)	13.97	0.94	–	14.88	<0.001	–
	Gender	−0.16	0.27	−0.02	−0.6	0.547	0.94
	Age	0.01	0.01	0.02	0.86	0.387	0.75
	Residence	−0.06	0.1	−0.02	−0.64	0.525	0.97
	Education	0.18	0.18	0.03	1	0.315	0.79
	**Relationship status**	**0.82**	**0.25**	**0.08**	**3.28**	**0.001**	**0.93**
	**DERS-8 Total score**	**−0.31**	**0.02**	**−0.47**	**−18.11**	**<0.001**	**0.89**

*Note*. Step 1: *F*(5, 1285) = 12.29, *p* < 0.001, adjusted R^2^ = 4.19%. Step 2: *F*(6, 1284) = 67.54, *p* < 0.001, adjusted R^2^ = 23.63%. Gender was coded as following: 1 = females, 2 = males. Residence was coded as following: 1 = villages, 2 = small towns (up to 20,000), 3 = towns (from 20,000 to 100,000), 4 = large cities (above 100,000 inhabitants). Education was coded as following: 1 = primary, 2 = vocational, 3 = secondary, 4 = higher. Relationship status was coded as following: 1 = single, 2 = in relationships. Statistically significant predictors are in bold. In our regression analysis, we controlled for gender, and for interpretability reasons data on 36 non-binary and 2 people with unidentifiable gender were excluded from the analysis; therefore, in this analysis, our sample comprised females and males only and it consisted of 1291 people.

## Data Availability

The raw data supporting the conclusions of this article will be made available by the authors on request.

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
