# Peer review of "A Screening Measure of Emotion Regulation Difficulties: Polish Norms and Psychometrics of the Difficulties in Emotion Regulation Scale-8 (DERS-8)"

_healthcare, 2025, doi:10.3390/healthcare13040432_

Round 1

Reviewer 1 Report

Comments and Suggestions for Authors

1. Retitle to be "Exploring Emotional Challenges in Poland: Psychometrics of the Emotion Regulation Scale (DERS-8) and Its Norms"

2. Recheck keywords into five words from Mesh Browser https://meshb.nlm.nih.gov/

3. From Line 79, the authors should indicate the primary and secondary objectives of the research instead of hypothesis declaration.

4. What benefit do you get after translating and validating DERS-8 in the Polish version? Please add them in the last paragraph of the introduction.

5. How do you calculate the number of participants? Please show with the reference.

6. Please report the participants' inclusion and exclusion criteria to ensure they are "general population."

7. Please give the validation of PHQ-4 and WHO-5 in the Polish version.

8. Table 6 is not necessary; you could add the confirmatory factor analysis results on each model in Figure 1

9. The discussion is too lengthy. Please revise all of this part. The first paragraph should confirm the results of each objective, compared with the original one you interpreted.

10. The authors should discuss using this measurement for clinical implications and policy, such as mental health in higher education or during the psychological interventions in some mood difficulties, such as people at risk of suicide; please see and cite Jatchavala, C. (2023). Interventions during the copycat suicide crisis among Thai students: A follow-up study. Journal of Medical Society37(2), 68-75.

11. Please update the reference to 2014-2024, except for the validation of measurement and classic model.

Author Response

We would like to thank the editor and the five reviewers for their positive and encouraging feedback on our submission. The constructive comments of reviewers helped us to significantly improve the quality of this submission. We have been through all comments one by one, edited the manuscript in detail, and added new material where required. We hope the editor and reviewers find the revised version of the manuscript clear and suitable for publication in Healthcare. All changes made are highlighted in red (in the replies and the revised paper).

Reviewer 1

1. Retitle to be "Exploring Emotional Challenges in Poland: Psychometrics of the Emotion Regulation Scale (DERS-8) and Its Norms".

Reply: We have considered your suggestion regarding changing the title. As the original name of the validated scale is the Difficulties in Emotion Regulation Scale-8 (DERS-8), we believe that we cannot modify the original name of the scale. As we developed norms for the Polish DERS-8 in the paper, we feel that mentioning "norms" in the title is relevant. In our title, we also want to highlight the screening nature of the DERS-8, which is highly relevant for healthcare practices. Therefore, based on these justifications, we would like to keep our current title: A Screening Measure of Emotion Regulation Difficulties: Polish Norms and Psychometrics of the Difficulties in Emotion Regulation Scale-8 (DERS-8). However, if necessary, we would be happy to revise this element further according to reviewer's and Editorial Board's requests.

2. Recheck keywords into five words from Mesh Browser https://meshb.nlm.nih.gov/

Reply: We have checked keywords in MeSH Browser, and they are relevant. There are 10 keywords in our paper, as it is according to the requirements of the Healthcare journal, which state as following: "Keywords: Three to ten pertinent keywords need to be added after the abstract. We recommend that the keywords are specific to the article, yet reasonably common within the subject discipline". Please see https://www.mdpi.com/journal/healthcare/instructions

3. From Line 79, the authors should indicate the primary and secondary objectives of the research instead of hypothesis declaration.

Reply: We have considered your suggestion, and we have specified the aim of the study. Now, it is one aim without primary and secondary objectives. The aim as it states in the paper, we also present here:"Given the above, the aim of the current study was to verify the psychometric properties of the first Polish version of the DERS-8 and to develop its current Polish norms". Regarding hypothesis, we believe that they should not be removed as in our view presenting hypotheses is according to standard methodological conventions. However, if necessary, we would be happy to revise this element further according to reviewer's and Editorial Board's requests.

4. What benefit do you get after translating and validating DERS-8 in the Polish version? Please add them in the last paragraph of the introduction.

Reply: Thank you for your comment. We have now edited and added the requested information in the introduction:

"In response, the DERS-8 was developed to assess difficulties in emotion regulation in both adolescents and adults (Penner et al., 2022). This brief, 1-dimensional, 8-item self-report scale demonstrates strong psychometric properties (Penner et al., 2022) while offering advantages in terms of its practical application. The reduced number of items makes the DERS-8 especially valuable in both clinical settings and long-term research supporting targeted interventions for older adults and providing precise emotion difficulties screening (Espirito-Santo et al., 2023). Moreover, given their brevity, short scales are appropriate for studies where participants need to evaluate themselves and others multiple times (Gosling et al., 2003). Furthermore, in some cases (e.g., online-based research, where respondents are unlikely to spend much time on the website), researchers may encounter constraints that force them to choose between using a very short tool or not measuring at all (Gosling et al., 2003). Finally, the increasing pace of life, as well as specific research conditions, often forces researchers to work with progressively shorter tools, leading to an increasing need for such measures (Rammstedt & John, 2007).

Emotion regulation difficulties are suggested to be a transdiagnostic risk factor for a wide range of mental health issues (Morawetz & Basten, 2024). In Poland, the prevalence of such issues, including high levels of anxiety and depression symptoms as well as decreased levels of well-being, is high (Larionow, 2023; Larionow & Mudło-Głagolska, 2023). Therefore, a brief and effective psychometric tool of emotion regulation difficulties is of high relevance for research and clinical practice, chiefly a screening and prevention of mental health problems. As such, we deemed that the DERS-8 was a good candidate as a screening measure of emotion regulation difficulties in a Polish sample."

The last sentence of the introduction: "Making implications for the assessment and treatment of emotion regulation issues more accessible, we strove to present Polish adult norms for the DERS-8. To help facilitate the interpretation of the scale scores in research and practice, we presented a guide of their interpretation”.

5. How do you calculate the number of participants? Please show with the reference.

Reply: Thank you for this suggestion. We have now added a requested piece of information with the reference into the paper: "For factor analytic studies, a sample size of more than 1000 participants is generally treated as excellent (Mundfrom et al., 2005). As such, our sample size was appropriate for examination of the DERS-8".

The added reference: Mundfrom, D.J.; Shaw, D.G; Ke, T.L. Minimum Sample Size Recommendations for Conducting Factor Analyses. Int. J. Test. 2005, 5(2), 159–168. https://doi.org/10.1207/s15327574ijt0502_4

6. Please report the participants' inclusion and exclusion criteria to ensure they are "general population."

Reply: As requested, we have now added these criteria in the paper: "The participants were people from a Polish adult population, without any specific features that were targeted during the recruitment process. Inclusion criteria were Polish-speaking people and an age of 18 years and above, who signed their informed consent for study participation. Exclusion criterion was inattentive responding (such respondents failed an attention check question that requested them to select a specific answer)".

7. Please give the validation of PHQ-4 and WHO-5 in the Polish version.

Reply: The PHQ-4 and WHO-5 have been previously validated in Polish. We have now indicated in the paper that "In this study, all questionnaires were used in their Polish versions, which previously demonstrated strong psychometric properties. Internal consistency reliability coefficients obtained in the current study were good, with omega and alpha coefficients of ≥0.79 (see Section 3.1 for details)". We have now also described several details about the validation of the PHQ-4 and WHO-5 in Section 2.1: "The Polish PHQ-4 has previously demonstrated strong psychometric properties, including an empirically supported 2-factor structure and good internal consistency reliability of ≥0.73 in a general community sample of Polish females and males (Larionow & MudÅ‚o-GÅ‚agolska, 2023)" and "For instance, this Polish tool was characterized by an empirically supported 1-factor structure and internal consistency reliability of 0.85 in a general community sample of Poles (Larionow, 2023)".

8. Table 6 is not necessary; you could add the confirmatory factor analysis results on each model in Figure 1.

Reply: We have considered your comment and after consultations we deem that moving the factor analysis results to Figure 1 will lead to the decreased readability. Factor analysis results are quite extensive, with a lot of fit indices and their corresponding values, which would make the figure overwhelming. Figures are usually used as supplements to the main text in order to visually demonstrate the data, while the main data are presented in the text. Following these arguments, we would like to keep our Table 6 as it is. However, if necessary, we would be happy to revise this element further according to reviewer's and Editorial Board's requests.

9. The discussion is too lengthy. Please revise all of this part. The first paragraph should confirm the results of each objective, compared with the original one you interpreted.

Reply: As we decided to present only one aim/objective of the study (not several objectives), we have shortly summarized the main results in the first paragraph of the discussion. Our results were very similar to the original English DERS-8, and we have now expressed this in the following sentences in the text: "The aim of the study was to examine the psychometric properties of the Polish version of the DERS-8 and to present its norms. Overall, much like the original English version (Penner et al., 2022), the Polish DERS-8 showed strong psychometric performance, including factorial validity, internal consistency reliability, convergent, divergent and discriminant validity”. Then, we discussed our results in subsequent subsections of the discussion in detail, including detailed comparisons with the results of the original validation study.

As our study has described a lot of psychometric properties, including factor structure, measurement invariance across different demographic differences, internal consistency reliability, convergent and divergent validity, discriminant validity, the predictive role of DERS-8 scores in mental health outcomes, demographic differences with the development of Polish norms as well as such standard section like practical implications, limitations, future directions and conclusions, we believe that in general our discussion cannot be short and it should be rather long if to discuss all the elements. We can agree that in general the discussion can be considered lengthy compared to other empirical papers, however, taking into account all the above-described elements of the discussion, we believe that its volume is appropriate.

10. The authors should discuss using this measurement for clinical implications and policy, such as mental health in higher education or during the psychological interventions in some mood difficulties, such as people at risk of suicide; please see and cite Jatchavala, C. (2023). Interventions during the copycat suicide crisis among Thai students: A follow-up study. Journal of Medical Society37(2), 68-75.

Reply: As suggested, we have now elaborated more on clinical implications and policy, including mental health in higher education or during the psychological interventions in some mood difficulties. Please see specific section 4.5. Norms and Their Interpretation and Other Practical Implications and the text:

"The development and implementation of interventions targeting emotion regulation in children and adults requires the use of well-operationalized measures of emotion regulation, with the DERS being a popular measure of interest (Lancastle et al., 2024). In the educational context, following the ideas that emotion regulation difficulties are positively related to academic burnout (Iuga & David, 2024) and as well student engagement and relations with peers and teachers (De Neve et al., 2023), we believe that the DERS-8 can be considered a good measure for quick screening of emotion regulation difficulties among students. In Poland, university students experienced high levels of anxiety and depression symptoms (Pawlicka et al., 2024), therefore screening assessment of emotion regulation difficulties and the development of psychological support programmes targeting psychological distress and emotion regulation abilities seem to be pertinent. With the brevity of the DERS-8, such monitoring is very accessible.

Additionally, the DERS-8 supports long-term research aimed at targeted interventions for older adults (Espirito-Santo et al., 2023). Moreover, the DERS-8 is also well-suited for studies requiring the tracking of emotion regulation difficulties over time or simultaneous evaluation of many constructs".

Thank you for your advice to read the paper by Jatchavala (2023). We have analyzed the suggested paper, which objective was "to survey patterns of psychotropic drug prescription, psychotherapy, and their associations with suicidal risk among Thai university students during the copycat suicide crisis and the subsequent 6 month follow-up". While we recognize the role of psychotropic drug prescription, psychotherapy in suicidal risk among Thai university students during the copycat suicide crisis, we could not find significant thoeretical and empirical connections of this article to the topic of our article. Therefore, at this moment, we believe that we could not cite the article by Jatchavala (2023). However, if necessary, we would be happy to revise this element further according to reviewer's and Editorial Board's requests.

11. Please update the reference to 2014-2024, except for the validation of measurement and classic model.

Reply: As suggested, we have now evaluated our references and made necessary changes in the references list.

Thank you for your time and helpful comments!

Reviewer 2 Report

Comments and Suggestions for Authors

1.      The abstract needs to be revised and formatted according to journal style

2.      Please replace forecasting with predicting “in forecasting psychopathology symptoms”

3.      Please justify the reasons to select DERS-8, describing used tools for assessing emotion regulation and then advantages of DERS-8

4.      Please provide statement about Permission to translation and validation of the DERS-8

5.      Please add date of the study

6.      Please provide duration of online data collection starting-ending date

7.      Please describe data management in terms of missing values, outliers, normal distribution of data, and accuracy of online data collection

8.      Please add DERS-8 scoring method and possible cut-off point of the scale  

Author Response

We would like to thank the editor and the five reviewers for their positive and encouraging feedback on our submission. The constructive comments of reviewers helped us to significantly improve the quality of this submission. We have been through all comments one by one, edited the manuscript in detail, and added new material where required. We hope the editor and reviewers find the revised version of the manuscript clear and suitable for publication in Healthcare. All changes made are highlighted in red (in the replies and the revised paper).

Reviewer 2

1.      The abstract needs to be revised and formatted according to journal style.

Reply: The abstract has now revised according to the journal's style. https://www.mdpi.com/journal/healthcare/instructions

2.      Please replace forecasting with predicting “in forecasting psychopathology symptoms”

Reply: As suggested, we have now replaced "forecasting" with "predicting".

3.      Please justify the reasons to select DERS-8, describing used tools for assessing emotion regulation and then advantages of DERS-8.

Reply: According to reviewer's request, in the introduction, you can find the revised and added information:

"The emotion regulation field is characterized by a variety of emotion regulation models and their corresponding measures. In a recent systematic review on emotion regulation models (Martínez-Priego et al., 2024), ten models were identified. The most prevalent model was the process model of emotion regulation by Gross (1998, 2015) with the Emotion Regulation Questionnaire (Gross & John, 2003) and the Process Model of Emotion Regulation Questionnaire (Olderbak et al., 2023) for measuring individual emotion regulation strategies (e.g., expressive suppression). The second most prevalent model was Gratz and Roemer's model of difficulties in emotion regulation. It identifies four key components of emotion regulation: (1) recognizing and understanding one's emotions, (2) accepting emotions, (3) managing behavior in alignment with goals while experiencing negative emotional states, and (4) using flexible, context-appropriate emotion regulation strategies to adjust emotional responses as needed to achieve personal goals and meet situational demands (Gratz & Roemer, 2004). Based on this model, the Difficulties in Emotion Regulation Scale (DERS) was developed (Gratz & Roemer, 2004).

The DERS is one of the commonly used tools for assessing difficulties in emotion regulation (for review, see Lancastle et al., 2024; Weiss et al., 2022). The original DERS-36 is a self-report questionnaire designed to assess six dimensions of difficulties in emotion regulation: (1) non-acceptance of emotional responses, (2) difficulties engaging in goal-directed behavior, (3) impulse control difficulties, (4) lack of emotional awareness, (5) limited access to emotion regulation strategies, and (6) lack of emotional clarity (Gratz & Roemer, 2004, p. 15–16; Hallion et al., 2018, p. 12). The DERS-36 is psychometrically reliable (Bjureberg et al., 2016) making it a cornerstone measure in both research and clinical contexts. Yet, with 36 items, the DERS-36 could be difficult to be applied in certain situations or environments, such as large-scale epidemiological research, or in clinical treatment context (Bjureberg et al., 2016). The previous evidence suggests that the DERS-36 performed similarly across gender (i.e., females vs males) and racial groups, supporting meaningful comparisons between demographic groupings (Gómez-Simón et al., 2014; Ritschel et al., 2015). Females tended to have greater difficulties in emotion regulation than males, however, the effect size of gender differences was small (e.g., Gómez-Simón et al., 2014; Gouveia et al., 2022).

In response, the DERS-8 was developed to assess difficulties in emotion regulation in both adolescents and adults (Penner et al., 2022). This brief, 1-dimensional, 8-item self-report scale demonstrates strong psychometric properties (Penner et al., 2022) while offering advantages in terms of its practical application. The reduced number of items makes the DERS-8 especially valuable in both clinical settings and long-term research supporting targeted interventions for older adults and providing precise emotion difficulties screening (Espirito-Santo et al., 2023). Moreover, given their brevity, short scales are appropriate for studies where participants need to evaluate themselves and others multiple times (Gosling et al., 2003). Furthermore, in some cases (e.g., online-based research, where respondents are unlikely to spend much time on the website), researchers may encounter constraints that force them to choose between using a very short tool or not measuring at all (Gosling et al., 2003). Finally, the increasing pace of life, as well as specific research conditions, often forces researchers to work with progressively shorter tools, leading to an increasing need for such measures (Rammstedt & John, 2007).

Emotion regulation difficulties are suggested to be a transdiagnostic risk factor for a wide range of mental health issues (Morawetz & Basten, 2024). In Poland, the prevalence of such issues, including high levels of anxiety and depression symptoms as well as decreased levels of well-being, is high (Larionow, 2023; Larionow & Mudło-Głagolska, 2023). Therefore, a brief and effective psychometric tool of emotion regulation difficulties is of high relevance for research and clinical practice, chiefly a screening and prevention of mental health problems. As such, we deemed that the DERS-8 was a good candidate as a screening measure of emotion regulation difficulties in a Polish sample".

4.      Please provide statement about Permission to translation and validation of the DERS-8.

Reply: As suggested, we have now provided this statement in the paper: "L. Steinberg, one of the authors of the original DERS-8 (Penner et al., 2022), granted us the permission to translate and validate the scale in Polish".

5.      Please add date of the study

Reply: As suggested, we have now added the date of the study in the paper in this sentence: "This research project was conducted in accordance with the Declaration of Helsinki Ethical Principles and was approved by the Ethics Committee of the Faculty of Psychology at Kazimierz Wielki University (No. 1/13.06.2022, with its latest revision: No. 3/11.11.2024)" and "From November to December in 2024, participants were invited to complete our online anonymous survey on social media platforms Facebook and Instagram, where we posted a link to the study with an appended consent form".

6.      Please provide duration of online data collection starting-ending date

Reply: As suggested, we have now added the date of the study in the paper in this sentence: "This research project was conducted in accordance with the Declaration of Helsinki Ethical Principles and was approved by the Ethics Committee of the Faculty of Psychology at Kazimierz Wielki University (No. 1/13.06.2022, with its latest revision: No. 3/11.11.2024)" and "From November to December in 2024, participants were invited to complete our online anonymous survey on social media platforms Facebook and Instagram, where we posted a link to the study with an appended consent form".

7.      Please describe data management in terms of missing values, outliers, normal distribution of data, and accuracy of online data collection

In the paper, we have now indicated that "There were no missing data, as replies on all questionnaires were mandatory".

As for normal distribution of data, we have now added the sentence in the paper: "All the study variables were reasonably normally distributed in the total sample (n = 1329), with skewness values from 0.12 to 0.55, and kurtosis values from -1.11 to -0.34".

As for outliers, we have now added the sentence in the paper: "Influential cases (outliers) were assessed using Cook's distance with a cut-off of 1. No outliers were detected".

As for accuracy of online data collection, we used an attention check question, and have now added a sentence: "Exclusion criterion was inattentive responding (such respondents failed an attention check question that requested them to select a specific answer)".

8.      Please add DERS-8 scoring method and possible cut-off point of the scale.

Reply: The information about the scoring is presented in the paper: "The total DERS-8 score is obtained by summing the individual item scores, with higher scores indicating more pronounced difficulties with emotion regulation". Scoring instructions are also presented in Supplementary Files. As for possible cut-off scores, please see the revised information in discussion: "To facilitate the interpretation of the DERS-8 scores, we proposed to classify levels of emotion regulation difficulties based on the following categories of percentile rank norms (Flanagan & Caltabiano, 2004): low levels (percentile ranks of ≤15), average levels (percentile ranks from 16 to 84) and high levels (percentile ranks of ≥85). Overall, a total DERS-8 score of 32 suggested high levels of emotion regulation difficulties in the total sample. We recommend to use age-specific cut-off scores for high levels of these difficulties, with a total DERS-8 score of ≥33 in younger adults aged 18–29 and a total DERS-8 score of ≥29 in older adults aged 30–73. We would like to underline that these cut-off scores were calculated based on the current Polish sample, and therefore may not generalize to other cultures".

Thank you for your time and helpful comments!

Reviewer 3 Report

Comments and Suggestions for Authors

The idea of your article is interesting, my recommendations are the following:

Abstract

It would be recommended to differentiate the population by sex-gender. This variable is not only of high interest in terms of emotional regulation, but it is important to reflect it with respect to international policies.

It is advisable to put the acronym DERS-8 immediately after the name (line 15). 

It would be interesting not only to explain what internal consistency has been evaluated, but also what internal consistency (data of interest).

It would be important to try to adapt to the journal's standards. 323 words are presented when the pattern is"Abstract: Systematic reviews and original research articles should have a structured abstract of around 250 words.

It would be desirable to explain why the psychometric properties are strong by presenting concrete data. .60 is not the same as .90, for example.

 Introduction

It would be interesting to include research objectives, beyond the hypotheses.

It would be interesting to expand the search for data by contrasting by age, differences by sex, environments... More speaking of the contrast of a scale, it is important to see their behavior in other places or by population strata and then contrast.

Methods

 It is important to expose the validity/reliability of the instruments used. For example, Cronbach's alpha. Not only in the contrasted questionnaire, but in those used.

Results

It would be great to check if all the exposed tables are necessary, given that they are higher than average.

Discussion

It would be interesting to contrast with the original version, or geographically close samples.

Conclusion

- 

References

-

Author Response

We would like to thank the editor and the five reviewers for their positive and encouraging feedback on our submission. The constructive comments of reviewers helped us to significantly improve the quality of this submission. We have been through all comments one by one, edited the manuscript in detail, and added new material where required. We hope the editor and reviewers find the revised version of the manuscript clear and suitable for publication in Healthcare. All changes made are highlighted in red (in the replies and the revised paper).

Reviewer 3

The idea of your article is interesting, my recommendations are the following:

Abstract

It would be recommended to differentiate the population by sex-gender. This variable is not only of high interest in terms of emotional regulation, but it is important to reflect it with respect to international policies.

Reply: We have now indicated the specific numbers of participants across genders in the abstract.

It is advisable to put the acronym DERS-8 immediately after the name (line 15). 

Reply: The acronym DERS-8 was put immediately after the full name of the tool. Please see the text in the paper: "Abstract: Background/Objectives: Difficulties in emotion regulation (DER) serve as a transdiagnostic risk factor for a wide range of emotion-based psychopathologies, including anxiety and depression disorders. This study presents a report on the psychometrics of the 8-item Difficulties in Emotion Regulation Scale-8 (DERS-8) and the development of its Polish norms".

It would be interesting not only to explain what internal consistency has been evaluated, but also what internal consistency (data of interest).

Reply: In the abstract, as suggested, we have now indicated the specific value of internal consistency reliability obtained in this study: "Our empirical evidence supported strong psychometrics of the Polish DERS-8, including its good level of internal consistency reliability (i.e., 0.89) and strong validity".

It would be important to try to adapt to the journal's standards. 323 words are presented when the pattern is"Abstract: Systematic reviews and original research articles should have a structured abstract of around 250 words.

Reply: As suggested, we have now shortened our abstract. It has now around 250 words.

It would be desirable to explain why the psychometric properties are strong by presenting concrete data. .60 is not the same as .90, for example.

Reply: We have now added the specific value of internal consistency reliability into the abstract. In terms of the other elements, considering word limits, we think the addition of these details would mean that other vital parts of the narrative of the abstract would need to be excluded. Hence, we have presently not added in these elements to the abstract (as often psychometric papers do not include these elements). However, if the editorial team would like these added we can revise the abstract further and add them.

Introduction

It would be interesting to include research objectives, beyond the hypotheses.

Reply: We have clarified our research aim along with hypotheses. Please you can find this text in the end of the introduction:

"Given the above, the aim of the current study was to verify the psychometric properties of the first Polish version of the DERS-8 and to develop its current Polish norms. We hypothesized that:

1. The Polish DERS-8 would be characterized by a 1-factor structure, with a good fit in confirmatory factor analysis, and it would be invariant across different demographic categories as well as mental health outcomes levels;

2. The Polish DERS-8 would have good internal consistency reliability;

3. Higher difficulties in emotion regulation, measured with the Polish DERS-8, would be associated with higher levels of psychological distress (i.e., anxiety and depression symptoms) and lower levels of well-being, supporting convergent and divergent validity of the scale;

4. The Polish DERS-8 would show good discriminant validity, with its scores being statistically separable from psychological distress and well-being;

5. The Polish DERS-8 scores would be meaningful statistical predictors of anxiety and depression symptoms as well as psychological well-being, while controlling for sociodemographic variables.

Making implications for the assessment and treatment of emotion regulation issues more accessible, we strove to present Polish adult norms for the DERS-8. To help facilitate the interpretation of the scale scores in research and practice, we presented a guide of their interpretation".

It would be interesting to expand the search for data by contrasting by age, differences by sex, environments... More speaking of the contrast of a scale, it is important to see their behavior in other places or by population strata and then contrast.

Reply: In the introduction, we strove to indicate the psychometric properties of the serioes of the DERS measures in relation to sex and age, and racial differences. As suggested, please find the added text in the paper: "The previous evidence suggests that the DERS-36 performed similarly across gender (i.e., females vs males) and racial groups, supporting meaningful comparisons between demographic groupings (Gómez-Simón et al., 2014; Ritschel et al., 2015). Females tended to have greater difficulties in emotion regulation than males, however, the effect size of gender differences was small (e.g., Gómez-Simón et al., 2014; Gouveia et al., 2022)".

Methods

It is important to expose the validity/reliability of the instruments used. For example, Cronbach's alpha. Not only in the contrasted questionnaire, but in those used.

Reply: We have now indicated in the paper that "In this study, all questionnaires were used in their Polish versions, which previously demonstrated strong psychometric properties. Internal consistency reliability coefficients obtained in the current study were good, with omega and alpha coefficients of ≥0.79 (see Section 3.1 for details)". We have now also described several details about the validation of the PHQ-4 and WHO-5 in Section 2.1: "The Polish PHQ-4 has previously demonstrated strong psychometric properties, including an empirically supported 2-factor structure and good internal consistency reliability of ≥0.73 in a general community sample of Polish females and males (Larionow & MudÅ‚o-GÅ‚agolska, 2023)" and "For instance, this Polish tool was characterized by an empirically supported 1-factor structure and internal consistency reliability of 0.85 in a general community sample of Poles (Larionow, 2023)".

Results

It would be great to check if all the exposed tables are necessary, given that they are higher than average.

Reply: Thank your suggestion. We believe that all tables are necessary as they present all necessary information regarding the data and conducted analysis. Deleting this information means that this information should be presented in the text, and in our opinion this will decrease the readability. Our previous experiences with publishing articles with the MDPI suggests that the Graphical Team of MDPI is very professional and they will help us in formatting the size of the tables.

Discussion

It would be interesting to contrast with the original version, or geographically close samples.

Reply: We would be happy to contrast our results with other culturally similar samples. However, as the DERS-8 has been recently developed (2022), cross-cultural data on the DERS-8 are very limited. To the best our knowledge, only original Enlish version of the scale is presented in the literature, and our Polish study seems to be the second validation study. Despite the limited data, we have now elaborated on demographic comparisons of emotion regulation difficulties in past studies conducted in geographically close samples.

"Our study revealed several demographic patterns in emotion regulation difficulties. Females reported greater struggles with managing negative emotions compared to males, supporting previous empirical findings that emotional challenges may be more pronounced in females (Ricarte Trives et al., 2016; Anderson et al., 2016). However, the effect size of these gender differences was small consistently with the previous findings (Giromini et al., 2017).

In a series of ANCOVAs devoted to the demographic differences analysis, age was a significant covariate. Further exploration of the role of age in emotion regulation difficulties allowed us to note that difficulty regulating emotions was more prevalent among younger individuals than in the older ones in our data-set. These results are consistent with earlier research, suggesting that ageing is associated with more efficient and adaptive emotion regulation due to the accumulation of life experience (Charles, 2010; Urry & Gross, 2010). Our results indicate that younger individuals are at risk of psychopathology development, therefore assessing emotion regulation difficulties, which serve as a risk factor for a wide range of psychopathologies (Espenes et al., 2024; McLaughlin et al., 2011; Miu et al., 2022), is particularly relevant for this group.

We were also interested in discovering whether education and relationship status categories (as less studied demographic variables in the emotion regulation field) differentiated difficulties with emotion regulation. We found that people without a university degree experienced slightly more difficulties in emotion regulation, but the effect size of these differences was negligible. The relationship status did not differentiate the DERS-8 scores.

To date, to the best of our knowledge, there are no studies on the demographic differences in DERS-8 scores, therefore, we cannot contrast our results with the previous ones. However, based on the obtained results and the past work on the full DERS (e.g., Giromini et al., 2017), we may conclude that age and gender seem to be the most relevant demographic factors, which should be taken into account while studying challenges in emotion regulation".

Conclusion

-

References

-

Thank you for your time and helpful comments!

Reviewer 4 Report

Comments and Suggestions for Authors

The manuscript "A Screening Measure of Emotion Regulation Difficulties: Polish 2 Norms and Psychometrics of the Difficulties in Emotion Regulation Scale-8 (DERS-8)" is generally a well-developed validation article in the Polish population. Although the manuscript is well written, I feel there are a few points that could be improved to make the whole thing more powerful.

  1. The Authors use the statement that the study sample was selected from the general population. This falsely suggests that we are dealing with a sample representative of the general population. However, the data in Table 1 do not indicate this, given the large disparities in gender prevalence, place of residence (most Poles come from small towns and villages, not large urban agglomerations) and education (well, in Poland it is not yet so good that 10% of adults have a higher education). For a sample to be representative for the general population, random selection must occur. Unfortunately, in this case we are dealing with a conventional sample, invited to online research, which is a next limitation. Not all Poles use Facebook and Instagram, especially middle-aged and older people. Therefore, please correct this misleading entry about the "general population" in both the abstract and the methodological section, and add more restrictions to the limitations section related to the lack of representativeness not only in terms of gender, but also place of residence and education.
  2. The replication of this study is impossible, because it is unclear how participants were invited to the study. Did you use private "friends lists" on Facebook and Instagram? If so, how many people were involved in recruiting participants for the study? Or did you use publicly available groups? How did you send invitations? Did you have permission from moderators? Please list all the groups you worked with in recruitment.
  3. How many people refused to participate in the study? What were the inclusion/exclusion criteria? How many people were excluded? What was the response rate for this study?
  4. Have you monitored in any way mental disorders confirmed by specialists (psychiatrists, psychologists)? Do you know how many people have such diagnoses (what disorders?) among the examined people? If not, please add this issue to the limitation of the study section.
  5. I am very concerned about the lack of precision in the use of terms in this article. The concept of psychopathology is very broad and includes many potential disorders. However, you have only studied two: symptoms of anxiety and depression. Please withdraw the terms "psychopathology levels", "psychopathology symptoms" and "psychopathology factor" (Figure 1, Model 1) from the entire article (from the abstract to the conclusions). This is very misleading, as it introduces confusion, suggesting that you have measured all possible symptoms of psychopathology, including psychiatric and behavioral disorders. Please replace this term with "distress" in each case in the whole manuscript, or simply use the terms "anxiety and depression", which is consistent with the facts.
  6. To confirm convergent and discriminant validity, it is now common to use the Composite CFA, with AVE and HTMT assessment. Please consider introducing these statistics, which would significantly enhance the strength of this manuscript.

Author Response

We would like to thank the editor and the five reviewers for their positive and encouraging feedback on our submission. The constructive comments of reviewers helped us to significantly improve the quality of this submission. We have been through all comments one by one, edited the manuscript in detail, and added new material where required. We hope the editor and reviewers find the revised version of the manuscript clear and suitable for publication in Healthcare. All changes made are highlighted in red (in the replies and the revised paper).

Reviewer 4

The manuscript "A Screening Measure of Emotion Regulation Difficulties: Polish 2 Norms and Psychometrics of the Difficulties in Emotion Regulation Scale-8 (DERS-8)" is generally a well-developed validation article in the Polish population. Although the manuscript is well written, I feel there are a few points that could be improved to make the whole thing more powerful.

1. The Authors use the statement that the study sample was selected from the general population. This falsely suggests that we are dealing with a sample representative of the general population. However, the data in Table 1 do not indicate this, given the large disparities in gender prevalence, place of residence (most Poles come from small towns and villages, not large urban agglomerations) and education (well, in Poland it is not yet so good that 10% of adults have a higher education). For a sample to be representative for the general population, random selection must occur. Unfortunately, in this case we are dealing with a conventional sample, invited to online research, which is a next limitation. Not all Poles use Facebook and Instagram, especially middle-aged and older people. Therefore, please correct this misleading entry about the "general population" in both the abstract and the methodological section, and add more restrictions to the limitations section related to the lack of representativeness not only in terms of gender, but also place of residence and education.

Reply: Thank you for your comments. We have considered your comment, and we feel that we use a term "a general population" without thinking of it as "a representative sample" or "a representative population" (we did not use the term "representative" in the paper). In our view, the terms "general population" and "representative population" are similar but different terms. In our paper, we do not use the term "representative" in relation to our sample. Our sample is not representative in respect to the demographic structure of the Polish population. Some authors argue that every sample is not representative, as it composed of people who have demonstrated willingness to participate in the study and signed informed consent, therefore, no people who have not signed the consent are included in that sample. In order to avoid any confusion related to "general" and "representative" populations/samples, we have now eliminated the terms "general sample/population" from the paper when describing the study sample, and we have now added a sentence which indicated that the study sample was not representative (e.g., the presence of gender imbalance).

In terms of other demographic characteristics like residence and education, we feel that there is no serious imbalance. We feel that there is a kind of misunderstanding regarding "place of residence (most Poles come from small towns and villages, not large urban agglomerations) and education (well, in Poland it is not yet so good that 10% of adults have a higher education)". In our paper, we did not report that "10% of adults have a higher education". Our sample was relatively diverse regarding the place of the residence and education (for details, please see Table 1). Therefore, we cannot say that the sample was not diverse in this regard. In general, as we did not say previously that the sample was representative and we do not treat our sample as a representative sample, therefore, justifications of sample representativeness cannot be applicable in this case. However, we have now also elaborated on the limitations related to the gender imbalance and the specificity of the sample.

Please find the text in the paper: "This study has several limitations. First, its cross-sectional design does not allow for determining whether emotion regulation difficulties lead to mental health issues or vice versa, highlighting the need for further investigation of causality through longitudinal research. Secondly, our sample was not fully representative of the structure of the entire Polish population (e.g., the presence of gender imbalance). Also, participants choose to engage in psychological studies that align with their personal needs and traits, potentially introducing an unintentional self-selection bias (Kaźmierczak et al., 2023). Additionally, online recruitment could have excluded individuals with limited internet access, potentially leading to underrepresentation of certain groups (e.g., older individuals and people with limited Internet access). Thirdly, the small number of non-binary participants limited the ability to explore gender-based differences fully between females, males, and non-binary individuals as well as to conduct measurement invariance across these three gender groups. Finally, by design, this was not a clinical study, therefore, we did not control participants' clinical diagnoses if they were any. Our study focused on a non-clinical adult sample, offering a foundation for future research involving more diverse groups, such as adolescents and individuals with various clinical diagnoses".

2. The replication of this study is impossible, because it is unclear how participants were invited to the study. Did you use private "friends lists" on Facebook and Instagram? If so, how many people were involved in recruiting participants for the study? Or did you use publicly available groups? How did you send invitations? Did you have permission from moderators? Please list all the groups you worked with in recruitment.

Reply: The link with the invite was posted on researchers' profiles (We have now clarified this in the paper: "This link was distributed on the authors' profiles"), and as we did not use publicly available groups, therefore, we had no need to have permission from moderators. We do not know many people were involved in recruiting participants for the study as this was an online study and the Google Forms does not allow to see how many unique views are presented. We know the number of people who replied to the consent form. These were 1388 people, and hen their data were analyzed in respect to inclusion and exclusion criteria. After these procedures, 1329 individuals composed the final sample examined in our study.

3. How many people refused to participate in the study? What were the inclusion/exclusion criteria? How many people were excluded? What was the response rate for this study?

Reply: We are providing the replies to these questions: How many people refused to participate in the study? What was the response rate for this study? We cannot provide this information as Google Forms does not allow to see how many unique views are presented. We know the number of people who replied to the consent form. These were 1388 people, and hen their data were analyzed in respect to inclusion and exclusion criteria. After these procedures, 1329 individuals composed the final sample examined in our study.

What were the inclusion/exclusion criteria? Please see the reply below (this information is also presented in the paper):

"The participants were people from a Polish adult population, without any specific features that were targeted during the recruitment process. Inclusion criteria were Polish-speaking people and an age of 18 years and above, who signed their informed consent for study participation. Exclusion criterion was inattentive responding (such respondents failed an attention check question that requested them to select a specific answer). A total of 1388 people replied to the consent form, and the data of 59 people were considered invalid as specified by the inclusion and exclusion criteria. As such, the final sample comprised of 1329 people".

4. Have you monitored in any way mental disorders confirmed by specialists (psychiatrists, psychologists)? Do you know how many people have such diagnoses (what disorders?) among the examined people? If not, please add this issue to the limitation of the study section.

Reply: This was not a clinical study, therefore, we have no data about mental disorders confirmed by specialists. We have now indicated this as a limitation. Please find the text in the paper: "Finally, by design, this was not a clinical study, therefore, we did not control participants' clinical diagnoses if they were any. Our study focused on a non-clinical adult sample, offering a foundation for future research involving more diverse groups, such as adolescents and individuals with various clinical diagnoses".

5. I am very concerned about the lack of precision in the use of terms in this article. The concept of psychopathology is very broad and includes many potential disorders. However, you have only studied two: symptoms of anxiety and depression. Please withdraw the terms "psychopathology levels", "psychopathology symptoms" and "psychopathology factor" (Figure 1, Model 1) from the entire article (from the abstract to the conclusions). This is very misleading, as it introduces confusion, suggesting that you have measured all possible symptoms of psychopathology, including psychiatric and behavioral disorders. Please replace this term with "distress" in each case in the whole manuscript, or simply use the terms "anxiety and depression", which is consistent with the facts.

Reply: Thank you for your comments. We agree with them, and we have now edited the terms as suggested.

6. To confirm convergent and discriminant validity, it is now common to use the Composite CFA, with AVE and HTMT assessment. Please consider introducing these statistics, which would significantly enhance the strength of this manuscript.

Reply: We have now elaborated on these analyses, and added analyses to support the validity. Please find the text in the paper:

In the Methods: "Additionally, we evaluated the 95% CI for the estimated correlations between the constructs from confirmatory factor analysis. If the 95% CI for the estimated correlation between two constructs does not include 1.0, this supports evidence of discriminant validity for these two constructs (Cheung et al., 2024). Finally, heterotrait-monotrait (HTMT) ratio of correlations (with a threshold value of 0.85) and an average variance extracted (AVE) value (with a threshold value of 0.5) were used to examine discriminant validity (Cheung et al., 2024)".

In the Results: "Additionally, the 95% CI for the estimated correlations between all the pairs of the examined constructs did not include 1.0, supporting evidence of discriminant validity for the constructs. Finally, AVE values were 0.57, and values of HTMT ratio of correlations were 0.81, further supporting good discriminant validity of the four constructs, as measured with the DERS-8, two PHQ-4 subscale scores, and WHO-5".

Thank you for your time and helpful comments!

Reviewer 5 Report

Comments and Suggestions for Authors

The study observes through quantitative analysis the validity of the small-scale version of the DERS scale, which measures difficulties respectively abilities of people to regulate one's emotions within Polish population. The study is well rounded and based on sound research. I think it needs some editing and the clarification of "missing" in relation to the gender of the sample population.

Author Response

We would like to thank the editor and the five reviewers for their positive and encouraging feedback on our submission. The constructive comments of reviewers helped us to significantly improve the quality of this submission. We have been through all comments one by one, edited the manuscript in detail, and added new material where required. We hope the editor and reviewers find the revised version of the manuscript clear and suitable for publication in Healthcare. All changes made are highlighted in red (in the replies and the revised paper).

Reviewer 5

The study observes through quantitative analysis the validity of the small-scale version of the DERS scale, which measures difficulties respectively abilities of people to regulate one's emotions within Polish population. The study is well rounded and based on sound research. I think it needs some editing and the clarification of "missing" in relation to the gender of the sample population.

Reply: Thank you for your suggestion and request to clarify the word "missing" in relation to gender. "Missing" referred to participants who did not report their gender. We have now clarified this and replaced the word missing with "Not reported/unidentifiable".

Thank you for your time and helpful comments!

Round 2

Reviewer 4 Report

Comments and Suggestions for Authors

Dear Authors,

Thank you for taking all my suggestions into account, and for your excellent cooperation in improving the manuscript. I recommend the manuscript for publication. Congratulations!